



# Permafrost saline water and Early to Mid-Holocene permafrost aggradation in Svalbard

Dotan Rotem[1,2], Vladimir Lyakhovsky[3], Hanne Hvidtfeldt Christiansen[2], Yehudit Harlavan[3], Yishai Weinstein[1]

[1]Department of Geography and Environment, Bar-Ilan University, Ramat-Gan, 52900, Israel.
[2]Geology Department, the University Centre in Svalbard, UNIS, Longyearbyen 9170, Norway.
[3]Geological Survey of Israel, 32 Yesha'yahu Leibowitz, Jerusalem 9692100, Israel.

*Correspondence to*: Dotan Rotem (dotanrotem1969@gmail.com)

**Abstract.** Deglaciation in Svalbard was followed-up by seawater ingression and the deposition of marine (deltaic) sediments in fjord valleys, while elastic rebound resulted in fast land uplift and the exposure of
these sediment to the atmosphere, therefore the formation of epigenetic permafrost. This was then followed by the accumulation of aeolian sediments, which froze syngenetically. The permafrost was drilled in the east Adventdalen valley, Svalbard, 3-4 km from the maximum up-valley reach of post-deglaciation seawater ingression, and its ground ice was measured for chemistry. While ground ice in the syngenetic part is basically fresh the epigenetic part reveals a frozen fresh-saline water interface
(FSI), with chloride concentrations increasing from the top of the epigenetic part (depth of 5.5 m) to about 15% that of seawater at 11 m. We applied a one-dimensional freezing model in order to examine the rate of top-down permafrost aggradation, which could accommodate with the observed frozen FSI. The model examined permafrost development under different scenarios of mean average air temperature, water-freezing temperature and the degree of pore-water freezing. We found that even at the relatively
high temperatures of the Early to mid-Holocene, permafrost could aggrade quite fast, e.g. down to 15 to 33 m in 200 years, therefore allowing freezing of the fresh-saline water interface despite of the relatively fast rebound rate and the resultant increase in topographic gradients toward the sea. This could be aided by non-complete pore water freezing, which possibly lead to slightly faster aggradation, resulting in the freezing of the entire marine section at that location (23 m) within less than 200 years. We conclude that
freezing should have occurred immediately after the exposure of the marine sediment to atmospheric conditions.



## 1. Introduction

Cycles of global warming and cooling are well documented in the geological history (e.g., Imbrie et al., 1993; Benn & Evans, 2014; Arnscheidt & Rothman, 2020). During the Pleistocene, these cycles followed

northern hemisphere glaciation and deglaciation, which influenced both marine and land temperatures (Park et al., 2019). This also affected the extent of cryotic conditions in the periglacial environment (e.g. Murton, 2021), i.e., the distribution of permafrost, which currently covers 22% of the northern hemisphere land (Obu et al., 2019). While temperatures during the Holocene were significantly higher than during the Last Glacial period, the retreat of glaciers and the follow-up elastic rebound and exposure of new

land in the Arctic and the sub-Arctic environment allowed freezing and the aggradation of permafrost (e.g. Landvik et al., 1988). Nevertheless, the relatively high temperatures during the Early and the mid-Holocene warm period raise questions about the timing of initiation and the extent of this process (e.g. Landvik et al., 1988; Humlum, 2005).

In Svalbard (Fig. 1), the fast retreat of glaciers during the end of Late-Pleistocene into the beginning of

the Holocene resulted in the ingression of seawater in fjord valleys, which was followed by gradual uplifting and exposure due to elastic rebound. This resulted in epigenetic permafrost aggradation followed by the deposition of fluviatile and aeolian sediments and the formation of syngenetic permafrost during the last ca. 4 ka (Gilbert et al., 2018). In the present study, we use the presence of saline water in the epigenetic permafrost to constrain the timing of freezing.

Permafrost is a soil or rock, which has been below zero temperature for at least two consecutive years (French, 2017). While winter freezing of the ground is common in a large extent of land areas, the existence of permafrost and its aggradation depends on the annual energy balance between atmosphere and the land (Black, 1954). Accordingly, permafrost develops when the land heat loss during winter exceeds the gain during the summer for long enough time. This is controlled by both seasonal solar

radiation and the soil/rock thermal properties. Heat exchange between soil and the atmosphere is also strongly affected by land cover, whereby permafrost usually is not developed neither under the sea nor beneath warm-base glaciers (Waller et al., 2012). The extent and depth of permafrost can be significantly reduced by thick vegetation or snow cover (e.g. Grünberg et al., 2020).

During the Last Glacial Maximum (LGM), Barents Sea and the Svalbard area (Fig. 1) were covered by

one to three ice caps (Mangerud et al., 2002; Patton et al., 2017). Glacier retreat has been followed since by elastic rebound, which is well documented in Svalbard (Bondevic et al., 1995; Lonne & Nemec, 2004;





Sessford et al., 2015), with a land rise of up to 130 m in eastern Svalbard and 65 m in the western part of the archipelago (Forman, 2004). In western Svalbard, the locus of this study, research indicates a fast land rise of 19-15 mm y$^{-1}$ during Early to the mid-Holocene (11.7 – 8.2 ka BP), which decreased to 5 - 4 mm y$^{-1}$ toward the end of mid-Holocene (Salvigsen, 1984; Sessford et al., 2015) and ca. 1 mm y$^{-1}$ during the late Holocene (last 4 ka, e.g. Forman et al., 2004).

Land uplift and exposure is accompanied by the establishment of a surficial drainage system, as well as the development of a groundwater flow network, which strongly depends on the rate of permafrost deepening (Edmunds et al., 2001). The permeability of frozen soils is greatly reduced (Burt and Williams, 1976; Cochand et al., 2019), such that extensive permafrost prevents penetrating of surface water and recharging groundwater (McEwen & de Marsily, 1991). While in sporadic and discontinuous permafrost, groundwater flow is possible through non-frozen sections or taliks, flow is practically impossible through continuous permafrost land areas (Lemieux et al., 2008; Walvoord and Kurylyk, 2016), while it may be active in sub-permafrost zones.

According to the Ghyben-Hertzberg approximation (Bear & Dagan 1964; Verruijt 1968), depth from seawater level to the fresh-saline water interface should be about 1:40 of the groundwater head above sea level. This ratio increases with decreasing salinity of saline water. With typical Early to mid-Holocene rebound rates of 15 to 4.5 mm y$^{-1}$ (Sessford et al., 2015), and assuming that groundwater table (saturated conditions) followed the topography, the fresh-saline interface is expected to be pushed downwards as deep as 120 to 36 m respectively, within 200 years after exposure. Groundwater table of 1 m below surface would result in a delay of 100 to 200 years, but also in this case a sediment section of tens of meters will be completely flushed within several hundred years. This occurs unless sediment freezing practically halts flow in the subsurface.

In this paper, we test this hypothesis by checking ground ice chemistry in a permafrost core from Adventdalen, Svalbard, and by a 1-D numerical heat transfer model that simulates permafrost aggradation under various surface temperature conditions and degrees of freezing. We show that epigenetic permafrost formation had to start soon after exposure even under the Early to mid-Holocene warm conditions.



## 2. Study site

Adventdalen is a U-shaped glacier valley located in western Spitzbergen, Svalbard, centred on 78.110N, 16.180E (Fig. 1). During the last glacial, the valley was eroded to the basement, which was then covered by glacial deposits (Elverhøi et al., 1995). This was followed by deglaciation, which was completed ca. 10.5 ka BP (Mangerud et al., 1992; Svendsen & Mangerud, 1997; Lønne & Lyså 2005, Farnsworth et al, 2020). Deglaciation was followed by up-valley seawater ingression, up to 13.5 km from the current end of the fjord (Cable et al., 2018; Lonne & Nemec, 2004) and the deposition of foredelta and deltaic deposits. Elastic rebound resulted in the exposure of the eastern valley before 9.5 ka BP, which progressed down-valley, arriving at the current coastline location at about 4 ka (Gilbert et al., 2018). Exposed surface was first covered by fluviatile sediments, followed by aeolian deposits during 4-2 ka. Permafrost at Svalbard, both epigenetic and syngenetic, is continuous and is estimated to be >100 m thick in the valleys (Humlum, 2005). Active-layer thickness is commonly 60–100 cm (Christiansen and Humlum, 2003; Gilbert et al., 2018; Weinstein et al., 2019).

Mean annual air and sea surface temperatures during Early to the mid-Holocene were 2-4 °C higher than today, as suggested by marine molluscs shells (Mangerudn & Svendsen, 2018), lacustrine alkenons (van der Bilt et al., 2018), flora DNA (Alsos et al., 2016) and models incorporating physical and biological considerations (e.g. Park et al., 2019). Since the mid-Holocene, a continuous decline in mean annual air temperature (hereafter MAAT) is recorded, changing into fast temperature rise during the last several decades (e.g. Christiansen et al., 2013).

The study site, Adventdalen East (ADE), is located on a river terrace (78.1722° N 16.0613° E), 9.8 Km from the Adventfjorden at 23 m a.s.l (Fig. 1). Permafrost section (valley-fill sediments) at the ADE site (20+ m) consists of a syngenetic part from 1 to 5.5 m depth, which includes a shallow 1.5 m of syngenetic fine-grained aeolian deposits, underlain by 3 m of fluviatile sediments (mud and pebbles, ice-rich). This is underlain by back-delta, deltaic and foredelta sediments (5.5-17.5 m), which cover glacial sand deposits (17.5-20 m), (Gilbert et al., 2018). The study site emerged above seawater at 9.2 ka (Gilbert et al., 2018), exposing it to atmospheric conditions, which allowed the development of groundwater system on one hand and possibly the aggradation of permafrost on the other hand.





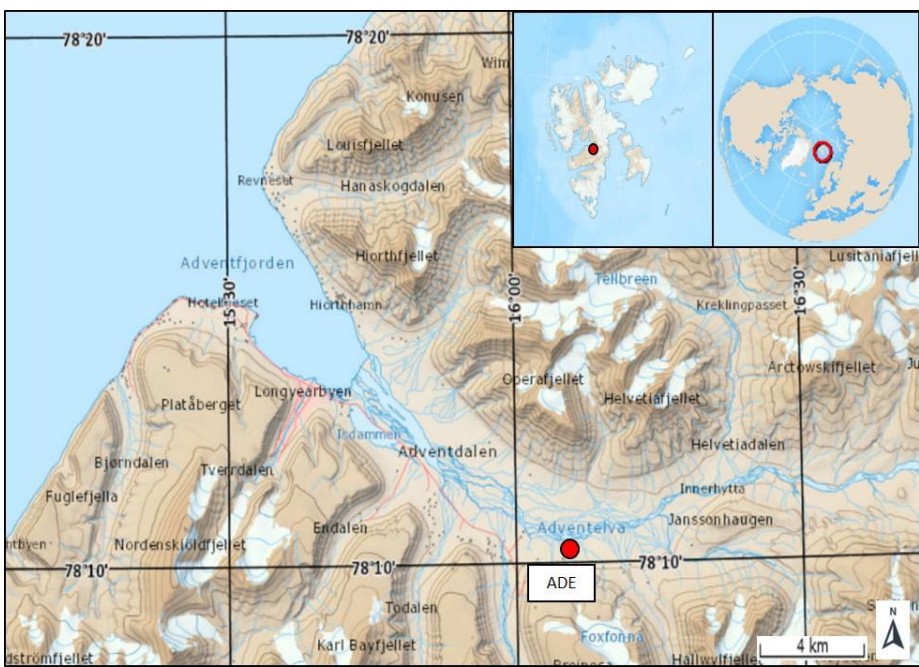

Figure 1. Study site ADE is located at the Adventdalen, Svalbard, on a river terrace. Map provided with courtesy of Norwegian Polar Institute.

## 3. Methods

Two cores, placed 0.5 m apart, one of 13 m and the other 9 m long, were retrieved at ADE in March 2017, using the UNIS permafrost drill-rig (Gilbert et al., 2015), which has barrels of 43 mm (ID). Core length, borehole depth, core condition, and gravel content were recorded in a field notebook. Retrieved core sections were placed in plastic bags and marked with serial number and an arrow pointing towards the core top. Cores were stored at -18°C in a freezer at UNIS until processing. Cores were sectioned in

a cold room (-5°C) to 0.5 m depth intervals. Intervals of the same depth in the two cores were combined in order to gain enough ground-ice per section for Ra isotope measurements. Samples were first scraped and then crashed to small chips, which were placed in 250 ml centrifuge tubes. Ra-free water (up to 40 ml) was added to some of the tubes, in order to facilitate the extraction of pore fluid. Samples were then thawed in a microwave set to 600 W for 2 min, followed by centrifuging for 8 min (11,000 RPM, high G)

in order to separate thawed water from the soil. Extracted water was run through 3 μm, followed by 0.45 μm filters. Most of the water was taken for Ra isotopes analysis (see Weinstein et al., 2019), while 30-60 ml was taken for chemistry analysis. Water of the added Ra-free water was analysed to correct for





element concentrations. Major elements were analysed in the Geological Survey of Israel (GSI) by ICP-AES (Optima 3000), where Sc was added as an internal standard, whereas $Cl^-$ and $SO_4^{2-}$ was determined

by potentiometer titration, using Metrohm 702 SM Titrino Titrator connected to a chlorine electrode. The error for all majors is considered less than 5%.

## 4. Chemistry

Major elements of thawed ground ice are presented in Table 1 and concentration profiles of $Cl^-$, $Na^{2+}$ and $SO_4^{-2}$ are shown in Fig. 2 a-c. While the salinity of ground ice in the syngenetic permafrost is that of

fresh water (e.g. $Cl^-$: 10-74 mg/L, $Na^{2+}$: 10-33 mg/L and $SO_4^{-2}$: 9-31 mg/L), epigenetic permafrost ground ice demonstrates a trend of increasing concentrations down to 9-12 m depth: 440-3600, 80-2700 and 150-740 mg/L of $Cl^-$, $Na^{2+}$ and $SO_4^{-2}$, respectively. Between 9-12 m, concentrations are quite scattered, and the increasing pattern is less clear. While the $Cl^-$ content (Fig. 2a) of the ground ice is no more than 15% seawater salinity, and the salinity of a deeper-seated saline water end member could be significantly

higher than seawater, the increased salinity clearly points to the location of afresh-saline water interface. The ionic ratio of $Na^{2+}$ to $Cl^-$ in both epigenetic and the syngenetic permafrost mostly exceeds 1 (Fig. 2d), significantly higher than in seawater (0.86), which is probably the result of sediment dissolution (e.g. of micas), since ion exchange should result in either a conservative behaviour (during freshening, as is the case in the ADE marine section) or in $Na^+$ depletion (in the case of salinization, e.g. Russak and

Sivan, 2010). On the other hand, $SO_4/Cl^-$ in the epigenetic permafrost is close to that of seawater (Fig. 2f), implying a relative conservative behaviour. Nevertheless, $SO_4/Cl^-$ in the syngenetic part is very variable (Fig. 2f), reaching ratios as high as 2, which could be due to shale dissolution (Hindshaw et al., 2016; Cabel et al., 2018). Higher than seawater ratios of what were also recorded in the upper 2 m of the epigenetic permafrost. High concentration of $SO_4$ was also recorded in sub-permafrost (Pingo) water,

which was attributed to gypsum dissolution (Hodson et al., 2020). Ca/Cl decreases with depth in the syngenetic permafrost and is very low (<0.01) in the epigenetic permafrost (Fig. 2e), which is in agreement with freshening experiments (Russak and Sivan 2010). The high ratio of Ca/Cl and $SO_4/Cl$ at -5.45 m depth is enigmatic and should be further studied.






Table 1. Major elements (mg/L) of grownd-ice samples of ADE core.

| Sample name | Depth (m) | Permafrost Type[1] | $Cl^-$ | $Br^-$ | $SO_4^{2-}$ | $SiO_2$ | $Na^+$ | $K^+$ | Sr | $Ca^{2+}$ | $Mg^{2+}$ |
|---|---|---|---|---|---|---|---|---|---|---|---|
| DR-AD-55 | -1.3 | Syngenetic | 27.1 | 0.5 | 19.0 | 13.0 | 10.7 | 17.0 | 0.2 | 5.4 | 15.1 |
| DR-AD-58 | -2.1 | Syngenetic | 12.8 | 0.6 | 30.7 | 14.0 | 19.8 | 4.1 | 0.4 | 8.8 | 27.8 |
| DR-AD-52 | -2.9 | Syngenetic | 74.1 | 0.7 | 8.9 | 9.0 | 13.6 | 44.7 | 0.6 | 12.7 | 35.2 |
| DR-AD-57 | -3.3 | Syngenetic | 13.9 | 0.5 | 27.9 | 5.9 | 32.6 | 4.4 | 0.1 | 3.6 | 7.6 |
| DR-AD-56 | -3.5 | Syngenetic | 10.0 | 0.3 | 12.3 | 5.3 | 13.3 | 4.5 | 0.1 | 1.5 | 2.6 |
| DR-AD-61 | -4.0 | Syngenetic | 14.5 | 0.7 | 13.0 | 11.6 | 9.5 | 5.2 | 0.2 | 3.6 | 8.8 |
| AD-DR-63 | -4.7 | Syngenetic | 26.8 | | 11.6 | 30.0 | 12.8 | 13.6 | 0.1 | 1.7 | 2.6 |
| DR-AD-59 | -5.5 | Epigenetic | 67.4 | 0.6 | 146.6 | 20.9 | 83.3 | 16.0 | 0.5 | 13.6 | 27.0 |
| DR-AD-53 | -6.3 | Epigenetic | 438.1 | 1.9 | 116.1 | 19.4 | 422.0 | 6.7 | 0.5 | 14.2 | 6.1 |
| AD-DR-64 | -7.1 | Epigenetic | 847.0 | | 369.5 | 38.1 | 828.7 | 22.7 | 0.1 | 11.6 | |
| AD-DR-65 | -8.3 | Epigenetic | 1678.4 | | 465.1 | 21.2 | 1296.3 | 29.1 | 0.1 | 0.5 | |
| AD-DR-67 | -8.8 | Epigenetic | 3677.6 | | 570.4 | 25.0 | 1680.1 | 55.6 | 0.1 | 2.2 | |
| DR-AD-54 | -9.4 | Epigenetic | 2145.1 | 7.9 | 508.2 | 9.1 | 2497.5 | 194.6 | 0.3 | 17.0 | |
| AD-DR-66 | -10.4 | Epigenetic | 3500.4 | | 738.9 | 24.8 | 2705.8 | 42.4 | 0.4 | 9.5 | |
| DR-AD-60 | -11.8 | Epigenetic | 2412.6 | 7.0 | 408.1 | 20.5 | 2319.8 | 98.0 | 0.6 | 18.3 | |

[1] after Gilbert et al., 2018




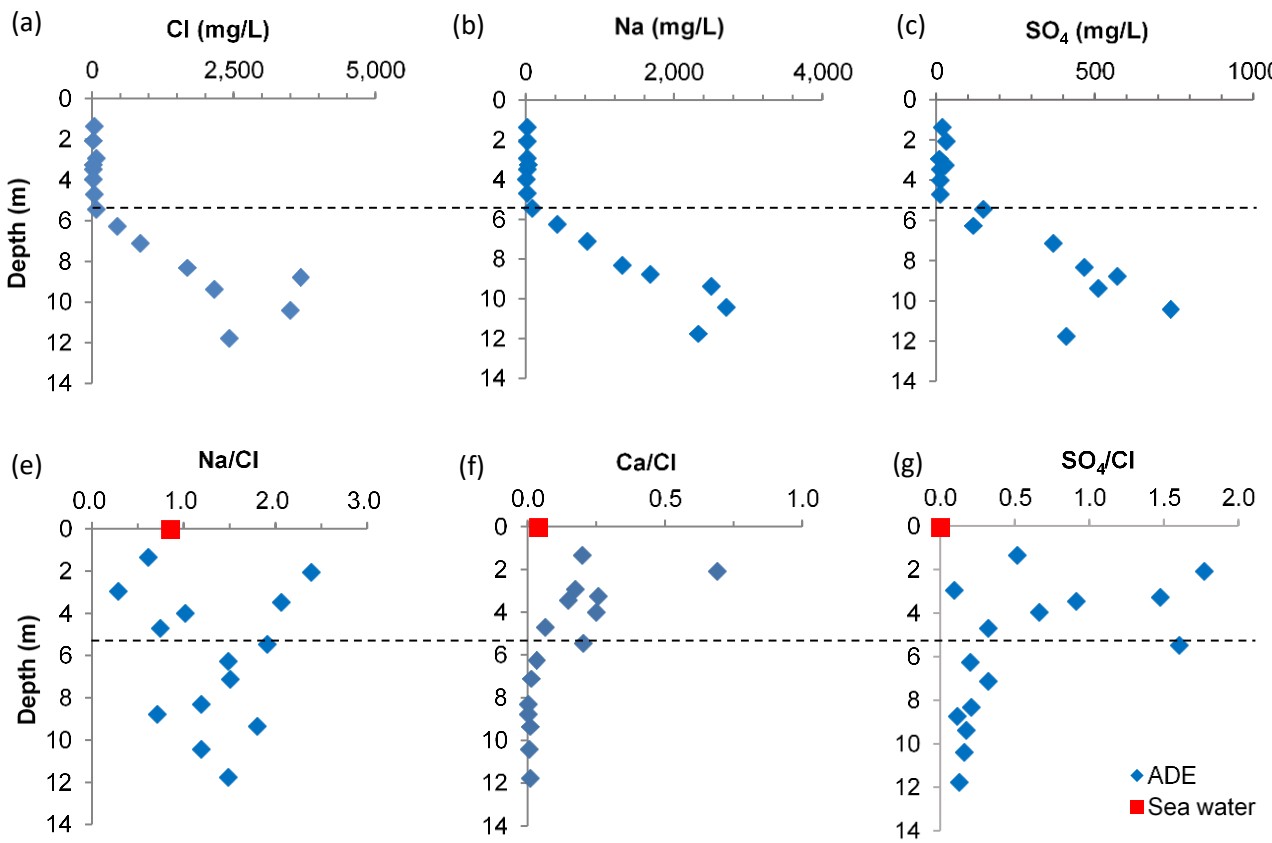

Figure 2. Major element concentrations in ground ice from Adventdalen-East (ADE) deep drillhole: (a) $Cl^-$, (b) $Na^+$ & (c) $SO_4^{2-}$ in mg/L. Figures (d) - (f) present selected equivalent ratios along the profile. Dashed line separates the syngenetic and the epigenetic permafrost (Gilbert et al., 2018).

## 5. Model of permafrost formation

### 5.1 Conceptual model

In order to study the possible rate of the permafrost formation we developed a numerical model that solves for the temperature distribution in space and time and freezing front progression, or Stefan





problem (e.g., Šarler, 1995). Considering low horizontal temperature variations, the problem is reduced

to one dimensional depth-dependent heat transfer with moving internal phase transition boundary. Various analytical and numerical methods have been developed to obtain stable solution of the Stefan problem (e.g., Crank, 1984). However, unlike the simple and clean single component systems, many natural systems, including water saturated porous rocks, change their phases under a specified temperature range rather than isothermally (Lunardini, 1987; Rühaak et al., 2015). In this case, evolving

"mushy zone" emerges and separates between the solid and liquid regions, where the thawing or freezing begins and proceeds, accompanied by latent heat absorbance or release (e.g., Crank, 1984; Yang et al., 2020). Following this approach, instead of the mathematical boundary, we use a narrow transition mushy zone (shaded area in Fig. 3), with pore space consisting of a mixture of ice and water (Rubinstein, 1982). The local enthalpy in the mushy zone takes values in the range between those of

the pure solid and liquid, and the temperature is approximated by a constant value, $T=T_f$, equal to the phase change temperature (Crank, 1984).

Heat exchange in the sub-surface is controlled by the ground temperature gradient, as well as by the soil thermal properties (i.e. thermal conductivity, Burn, 2011). Above ground, the main factor is the air temperature, which is measured and reported as MAAT (Luo et al., 2018; Szafraniec & Dobiński, 2020),

which is taken as representing the MAGST (Mean Annual Ground Surface Temperature). Initial surface temperature is defined according to the temperature of the shallow seawater during mid-Holocene (2°C, Rasmussen et al., 2012), while the initial temperature profile (Black line in Fig. 3) is defined using the regional average geothermal gradient of 0.033°C m$^{-1}$, as discussed by Olaussen et al., (2019) and Betlem et al., (2018). The lower boundary of the model was set at 300 m depth, with temperature of 12°C

according to the initial temperature distribution. Throughout the simulation, we search for the depth and time-dependent temperature distribution $T(z,t)$, schematically shown as a dashed curved line in Fig. 3. MAAT was set to several values: (1) the current -5.8°C (measured at the Adventdalenen 'Polygons' site, (Christiansen, 2005), 7 km from the fjord; (2) -4°C, which was taken from climate simulation models for the mid-Holocene (Park et al., 2019; see also Mangerud and Svendsen 2018; Van der Bilt et al., 2019);

(3) -3°C and 0°C assumed by Humlum (2005) for the mid-Holocene. While snow may cause differences between MAAT and MAGST due to thermal insulation (Zhang, 2005), it was found that in western Svalbard, specifically in flat landforms at Adventdalen, the differences between MAAT and MAGST are less than 0.5 °C (Christiansen, 2005; Lüthi 2010; Etzelmüller et al., 2011; Farnsworth 2013). Amplitude





of seasonal temperature oscillation at the surface was set to 12°C, similar to the current fluctuation (Nordli
et al., 2014; Chrisiansen, 2005; Osuch & Wawrzyniak, 2017).

In order to distinguish between a frozen and water saturated cell, we define a time and depth-dependent
freezing ratio, $B(z,t)$, shown by points line in Fig. 3. $B=1$ means that the soil is water-saturated, while in
the case of $B=0$ pore space is fully ice-saturated. In the mushy zone (shaded area in Fig. 3), the $B$-value
changes between 0 and 1, and the rate of its change defines the amount of energy or latent heat
associated with water-ice phase transition. Since it is now well-established that permafrost is not
necessarily fully frozen (e.g. Keating et al., 2017; Oldenborger & LeBlanc, 2018), we also investigated
permafrost aggradation under "partial freezing" conditions of 25% and 50%. We note that our model
assumes fully-saturated pore-water conditions, since freezing starts at sea level, soon after exposure,
therefore groundwater level is expected to be at the surface.

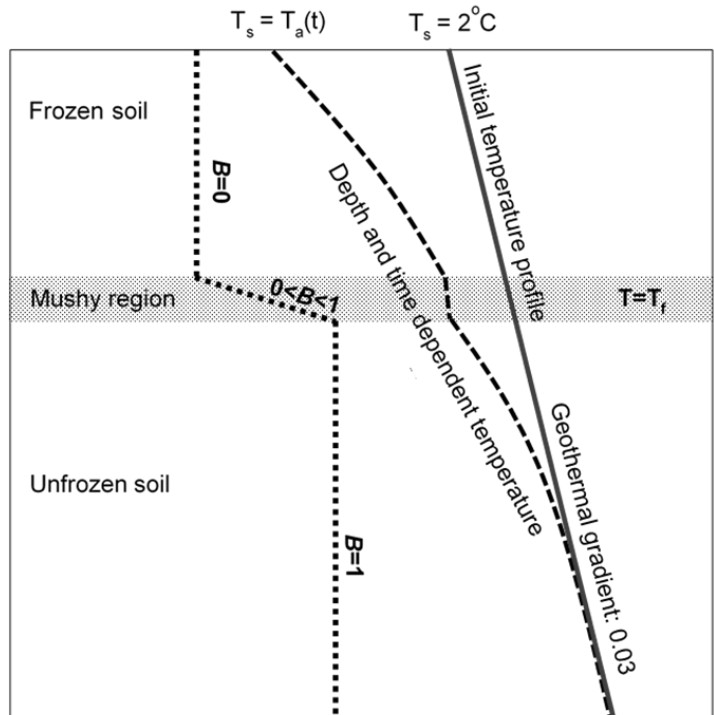

Figure 3. Schematic freezing profile during the top-down (epigenetic) freezing process. Initial and the
developing geothermal gradients are also shown; the mushy region with constant temperature is the
depth zone where phase transition occurs. Points line schematically represents the freezing condition,
where 0 stand for fully frozen and 1 for liquid only. Active layer seasonality is neglected.






Another factor affecting the rate of permafrost formation is the water-freezing temperature (hereafter: WFT), which varies with salinity (Morgenstern & Anderson, 1973), as well as due to other environmental factors (Farouki, 1981; Morgenstern & Anderson 1973). Since salinities in the ADE site, down to 12 m, do not exceed 15% that of seawater, fresh water freezing temperature is the most appropriate for our simulations. Nevertheless, we also conducted simulations with WFT of -2°C, which is close to that of seawater ($T_m$=-1.9°C, Marion et al., 1999; Bodnar, 1993), as well as with -6°C, following the reports of gilbert et al., (2019) and Tavakoli et al., (2021) about high salinities (up to 73 ppt) in Adventdalen permafrost, which may reduce the freezing temperature to -5° and -6°C, respectively

Porosity is an important factor in the aggradation or thawing of permafrost (Hornum et al., 2020). It determines both the amount of heat released or required during freezing/thawing, and the thermal characteristics of the soil (ice has significantly higher heat conductivity than water; Farouki, 1981). Porosity of sediments at ADE was mainly set to its average value, 0.3 (Gilbert et al., 2018), although we also tested the effect of other porosity values (Appendix 1). Bedrock (≥25 m depth) porosity was taken as 0.1, the value suggested by Hornum et al. (2020), assuming that the bedrock is mainly composed of fractured shales (Grundvåg et al., 2019; Benn and Evans, 2014).

Freezing temperature and degree of freezing were kept uniform in each of the simulations, although as pore water freezes, the remaining fluid becomes saltier, further lowering the freezing temperature of the remaining solute (Herut et al., 1990), such that fully frozen pore space (i.e. 100% freezing) can only be reached at extremely low temperatures, which are not relevant to our study sites, as well as to most other permafrost areas (Homshaw, 1980; Dobinski, 2011).

1-D freezing model provides a good approximation of freezing rate and permafrost aggradation as shown by many studies (e.g., Harada & Yoshikawa, 1996; Kukkonen & Šafanda, 2001; Farbrot et al., 2007; Etzelmüller et al., 2011; Hornum et al., 2020). Such models actually provide the maximum rates of freezing propagation as lateral heat transport by groundwater flow is neglected. Neglecting lateral heat transfer is quite justified, considering that (1) upstream shallow groundwater arrive from areas that were exposed earlier, therefore should not be warmer than the ADE groundwater, (2) experimental data suggest that temperature drop by tenth of centigrade below 0°C reduces hydraulic conductivity by several orders of magnitude (Burt & Williams, 1976; Rühaak et al., 2015). Nevertheless, as stressed above, soon after freezing initiates, hydraulic conductivity dramatically decreases and the impact of lateral flow could be neglected.



## 5.2 Mathematical formulation

The heat conduction equation for time and depth-dependent temperature profile, $T(z,t)$, is (Crank, 1984):

$$\rho Cp \frac{\partial T}{\partial t} = \frac{\partial}{\partial z}\left(\kappa \frac{\partial T}{\partial z}\right) + Q \tag{1}$$

where $Q$ is the energy sink or source, representing the latent heat associated with water-ice phase transition, $\rho$ is the soil density, $C_p$ is the specific heat capacity and $K$ is heat conductivity. The depth and time-dependent material properties were calculated assuming linear superposition of the soil, water, and ice properties (e.g., Lunardini, 1988). Thus, the depth-dependent density, heat capacity, and thermal

conductivity were calculated using the porosity, $\theta$, and freezing ratio, $B(z,t)$:

$$\rho = (1-\theta)\,\rho_{soil} + \theta\,((1-B)\rho_{ice} + B\rho_{water}))$$
$$\kappa = (1-\theta)\,\kappa_{soil} + \theta\,((1-B)\kappa_{ice} + B\kappa_{water})) \tag{2}$$
$$Cp = (1-\theta)\,Cp_{soil} + \theta\,((1-B)Cp_{ice} + BCp_{water}))$$


The thermal properties and density used for all system components (soil, ice and water) are listed in Table 2.

Out of the mushy zone, for the completely frozen (B=0) or unfrozen (B=1) sediments, the heat exchange leads to its cooling below or heating above the freezing temperature. When no latent heat is involved,

assuming homogeneous heat conductivity, the heat conduction equation (1) is reduced to:

$$\frac{\partial T}{\partial t} = D \frac{\partial^2 T}{\partial z^2} \tag{3}$$

where D (diffusivity) [m² s⁻¹] defined as:

$$D = \frac{\kappa}{\rho Cp}$$

In the mushy zone, where the water-ice phase transition occurs, both the $B$ value and the thermal

properties ($C_p$ and $K$) are depth-dependent. Accordingly, the complete heat conduction equation (1) is solved, including the latent heat term. The heat source/sink is equal to the mass of the freezing/thawing water per unit time multiplied by the latent heat, $L$. The water mass is equal to the rate of the $B$-value change times porosity and density. Finally, the source term is:

$$Q = -L\,\theta\rho\,\frac{\partial B}{\partial t} \tag{4}$$





We neglected the kinetics of the phase transition and assumed that thermodynamic equilibrium is established instantaneously in the mushy zone. This means that the rate of freezing/thawing is defined by the heat flux to and from the mushy zone with 0<B<1. Substituting (4) into (1) and using $\frac{\partial T}{\partial t} = 0$ leads to:

$$\theta \, L \, \rho \, \frac{\partial B}{\partial t} = \frac{\partial}{\partial z}\left(\kappa \frac{\partial T}{\partial z}\right) \tag{5}$$

The above equations are solved numerically for two functions $T(z,t)$ and $B(z,t)$, using the explicit-in-time finite difference scheme. These functions were approximated using the constant grid steps in depth $\Delta z$ and in time $\Delta t$:

$$T_{n,m} = T(n\Delta z, m\Delta t)$$

$$B_{n,m} = B(n\Delta z, m\Delta t)$$

where $n$ is the grid point number ($z = n\Delta z$ ) and $m$ is the time step number ($t = m\Delta t$ ). With this notation, the finite difference form of the heat conduction equation (1) is:

$$\rho_n \, Cp_n \, \frac{T_{n,m+1} - T_{n,m}}{\Delta t} =$$

$$= \frac{1}{\Delta z^2}\left[\frac{\kappa_{m+1}+\kappa_m}{2}(T_{n+1,m} - T_{n,m}) - \frac{\kappa_m+\kappa_{m-1}}{2}(T_{n,m} - T_{n,m-1})\right] + L\rho\theta \frac{B_{n,m+1}-B_{n,m}}{\Delta t} \tag{6}$$

where density and the thermal properties are calculated using equation (2).

Equation (6) is solved using time step $\Delta t = 32{,}600 \, s$ (0.5 day) and a depth spacing $\Delta z = 0.25 \, m$. The solution was obtained for the model size down to 300 m depth, summing up to 1,200 grid points. These numerical parameters satisfy the von Neumann stability condition for explicit-in-time numerical scheme (e.g., Ames, 1977) for the material properties of Table 2. The numerical code was written with Python (Wang & Oliphant, 2012). It allows simulating the permafrost dynamics and sub-surface sediments

freezing under various scenarios of MAAT, water freezing temperatures (WFT) and freezing extent of pore space water.

Table 2: 1-D heat transfer model physical parameters of water ice and soil.

| | Thermal conductivity | Heat capacity | Density | Diffusivity | Latent heat |
|---|---|---|---|---|---|
| | $\kappa$ (Wm/K) | Cp (J/K*Kg) | $\rho$ (Kg/m^3) | D (m$^2$/s)[*] | L (J/Kg) |



| | | | | | |
|---|---|---|---|---|---|
| Ice | | 2.24 | 2100 | 916.2 | 1.17X10-6 | |
| Water | | 0.569 | 4192 | 999.85 | 1.36X10-7 | 334000 |
| Soil (Silt) | dry unfrozen | 0.35 | 837 | 2400 | 1.74x10-7 unfrozen |
| | dry frozen | | 712 | | 2.04x10-7 frozen |

* D=κ /ρ*Cp


## 5.3 Model results

We present the results of model runs with variable combinations of surface temperature, water freezing temperature and sediments porosities. Complementary modelling results are presented in the Appendix. In all cases, simulations started in the spring (May) and followed an amplitude of 12°C around the chosen

(fixed) MAAT.

We first present simulations with MAAT of -4 ± 12°C, WFT 0°C, and complete (100%) freezing. Figure 4a presents results of a one-year simulation, with temperature profiles shown every second month. The model suggests that freezing under these conditions can reach down to two-meter depth within the first year. The freezing depth increases to 4 m within 6 years with a slight, but significant deepening of the

winter inflection point (Fig. 4b). After 50 years, freezing arrives at 10-11 m, and within 1000 years freezing front is already at 50 m (Fig. 4c) in the basement rocks (considering that sediment cover at ADE is ca. 25 m). The depth affected by cooling also progresses with time (<50 m in 50 years, >150 m in 1000 years) and T profile approaches linearity.



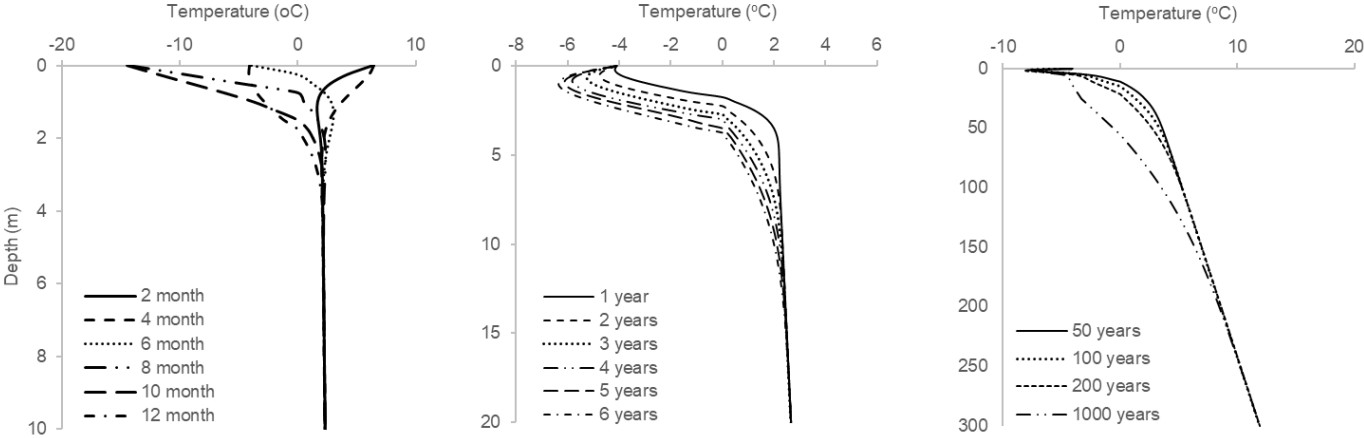


Figure 4. 1-D freezing model results for: (a) 1 year; the model starts (and finishes) at spring, defined as mid-time between minimum and maximum surface T (e.g. May), followed by summer increase in temperature (black and red lines), cooling and start of freezing during fall (grey line, e.g. November), followed by colder winter months (yellow and blue) and concluding in the spring (green); (b) 6 years

(starting in spring). MAAT -4 ± 12ºC, WFT set at 0ºC, 100% freezing. (C) Model simulation for 50 to 1000 years. Red dots denote -4ºC. Note the different scale between (a) (b) and (c).

Lowering the WFT results in decrease of the freezing rate. For example, with WFT of 0ºC freezing front will reach 15 and 21 m after 100 and 200 years while with WFT of -2ºC and -5ºC it will reach 11 and 16

m and 4.75 and 6.75 m during the same periods, respectively (Fig. 5). Surprisingly, permafrost aggrades even under WFT of -5ºC, which is lower than the annual average air temperature (MAAT) of -4ºC. This is because the thermal conductivity of ice is higher than that of water (Farouki, 1981), which results in a deeper advance of the winter freezing front (through ice) than the advance of summer thawing (through water).




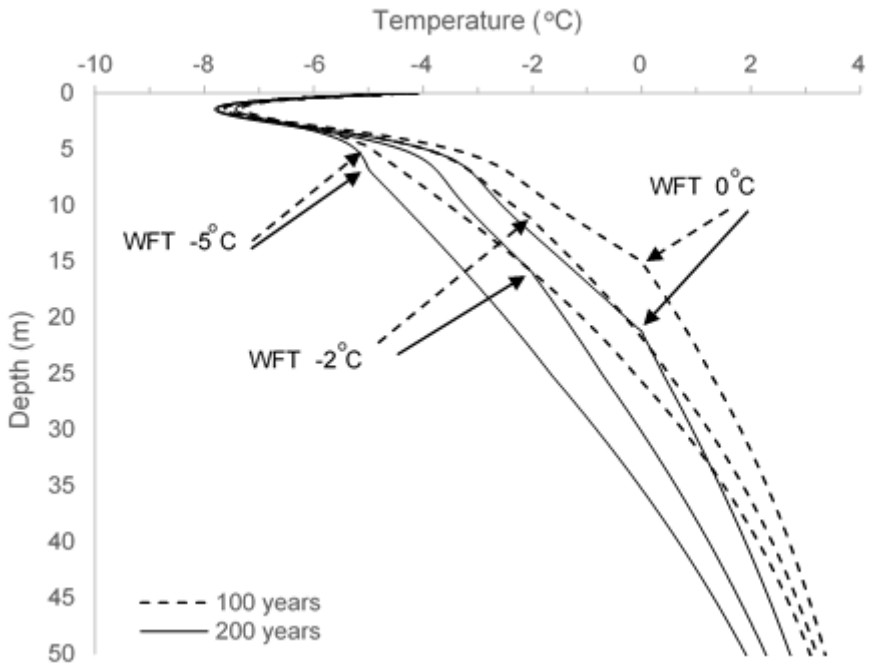

Figure 5. 1-D freezing model results with mid-Holocene MAAT of -4 °C ± 12 °C (Park et al., 2019), with
pore-water freezing temperature (WFT), taken as 0 °C -2 °C and -5 °C. Simulations include 100 and 200
years.

In Figure 6, we examine the effect of partial freezing. Partial freezing (25% and 50% in our scenarios)
result in deepening of the freezing depth, but differences are relatively small. With MAAT of -4 °C and
WFT of 0 °C, after 200 years freezing depth will reach 21, 22 and 25 m with 100% 50% and 25% freezing,
respectively. Lowering WFT to -2 °C the freezing depth will differ by less than 1 m (16 to 15.25 m), within
the three scenarios (Fig. 6). This is because of the trade-of between reducing latent heat and the lower
thermal conductivity of the partially frozen pore space. Moreover, with a WFT of -5°C, the trend changes,
and permafrost aggradation will occur only under the 100% freezing scenario. Under partial freezing of
25-50%, it will not exceed deeper than 2 m (Fig. 6), i.e. no permafrost will develop (assuming active layer
depth of 1-2 m), which suggests that when WFT is lower than the average MAAT (-4°C), aggradation is
controlled by the ice thermal conductivity rather than by latent heat. Last, with WFT of -6 °C (i.e.
significantly lower than the MAAT), there is no apparent permafrost aggradation also with 100% freezing





(assuming active layer depth of 2 m, Fig. 6), although permafrost does develop to ca. 3.5 m after 1000 year (not shown).


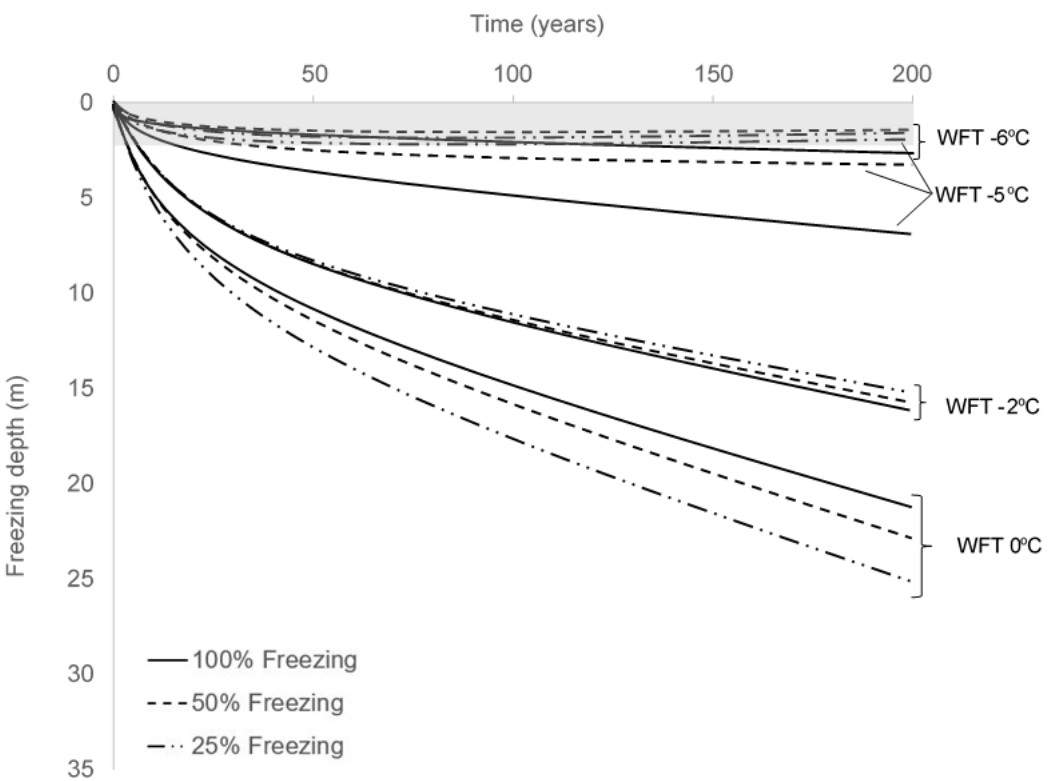

Figure 6. Mid Holocene permafrost aggradation with variable freezing degrees. MAAT was set to -4 °C (Park et al., 2019), and freezing proportions were taken as 100%, 50% and 25%. The shaded/grayish zone represents a hypothetical, conservative, active layer, which is taken at 2 m. We note that currently active layer at the area is ≤1 m, and 2 m was chosen due to the higher temperature during the mid-Holocene.

In general, higher MAAT leads to the lower aggradation rate (Fig. 7a-b). However, we show that even with MAAT > WFT (e.g., 0 °C and -2 °C, respectively) freezing will arrive at 3.5 m after 200 years and 8 m after 1000 years (not shown). As mentioned above, this is due to the asymmetry in the seasonal



freezing/thawing process. Higher conductivity during winter (frozen pore space) enhances the loss of heat, while the lower conductivity during summer (warming front goes through thawed pore space) slows the thawing process (Kukkonen & Šafanda, 2001). We note that lower proportions of freezing (e.g., 25%) will have the effect of reducing this asymmetry due to the higher proportions of liquid water in the cryotic
pore-space, therefore lower thermal conductivity during freezing. Accordingly, permafrost deepening is hardly observed in the scenario of MAAT=0 °C (in particular with WFT= -2 °C (compare Fig. 7a and 7b).

## 6. Discussion

### 6.1 Ground ice salinity and the frozen interface

When freezing front propagates downwards in a recently emerged land and epigenetic permafrost is formed, it might freeze old subsurface brines (Cascoyne, 2000). As freezing process proceeds, solute concentrations in the non-frozen residual water commonly increase (e.g., Cocks and Brower, 1974; Herut et al., 1990; El Kadi and Janajreh, 2017), which results in pore space with low water activity and high salt concentrations. The resulted brines may then migrate away from the freezing surface, driven by
density and capillary forces, and coalesce to form separate saline water lenses ('cryopegs'; Cascoyne, 2000). The level of salinity and water composition will depend on the initial water composition and the extent of freezing.

Complete permafrost freezing can hardly be obtained, since the eutectic point of seawater freezing is at -36°C to -54°C (Gitterman 1937; Ringer 1905; Nelson & Thompson 1954; Marion et al., 1999), while at
Adventdalen permafrost temperatures does not usually get below -6 °C (Christiansen et al., 2010) and are never lower than -12 °C even in the shallow permafrost (Christiansen et al., 2005; Isaksen et al., 2007). Although the eutectic point is well below the expected temperature values, freezing-salt expulsion process still prevails. Under these conditions, permafrost pore space should hold a small fraction of residual brine solution, which contains most of the solutes originally dissolved in the bulk pore-space
water. Partly unfrozen permafrost has been often observed in the study area, during drilling, in particular deeper than a few meters, where seasonal impact is not relevant. This was also deciphered from geophysical and geochemical observations (e.g., Keating et al., 2018; Weinstein et al., 2019). Nevertheless, when ground ice is thawed and collected, the extracted fluid from the relatively large segments should roughly indicate on the salinity of the original in situ pore fluid. Pore water composition





may be significantly altered from the original fluid that circulated in the sediments (e.g., seawater) due to ion exchange or dissolution prior to or even after cryotic conditions took place, which is reflectd in the Na/Cl and SO4/Cl  ratios in the thawed ground ice (Fig. 2). Nevertheles, Cl⁻ concentration is probably close to and represents the salinity of the original pore fluid. We note that while in certain cases permafrost contains lenses or pockets of unfrozen brine-containing cryotic soils ('cryopegs', e.g. Van

Everdingen, 1998), which are commonly attributed to the segregation and migration of fluids (i.e. non in situ), the relatively low salinity (Fig. 2) and the evident mixing profile (Fig. 2a, b and C), suggest that this is not the case in the ADE site, and that the observed fresh-seawater interface is an in situ observation. We relate to the chemistry of the extracted fluid as 'ground ice chemistry', although it could as well be that some of it was not actually frozen.

Cable et al., (2018) presented ground ice chemistry of cores from the Adventdalen, albeit closer to the current fjord (<4 km), west of the ADE site. In these cores, chloride, Sodium and Sulphate concentrations at depths of 3-11 m were up to 50% that of seawater. At ADE, farther away from the sea, ground ice in the epigenetic permafrost, from 5.5 m, shows a gradual increase in salinity (i.e., fresh-saline interface), with Cl⁻ concentrations reaching 15% that of seawater at 9 m below the surface. Although salinities do

not change much between 9-12 m, it is likely that a more saline water, close to seawater salinity, either exists today or existed in the past (prior to freezing) at deeper permafrost levels.

The existence of a fresh-saline interface in the very shallow permafrost suggests that freezing at ADE occurred straightaway after emergence above seawater. This is further discussed below.

**6.2 Rebound, exposure and fresh-saline interface deepening**

Assuming that Early Holocene (11-8 ka BP) precipitation was slightly higher than present (200 mm per year, Kjellman et al., 2020; McFarlin et al., 2018), and using a conservative infiltration factor of 0.2 and the porosity used in our simulations (0.3), this amounts to an annual rainfall infiltration of 120 mm per year. This could easily keep with the Early Holocene rebound rates of 15 mm y⁻¹ (established for the nearby Sassendalen Vallley, Salvigsen, 1984; Sessford et al., 2015), therefore preserve groundwater

table close to the surface of the emerging land. Using a Ghyben-Herzberg approximation (Bear & Dagan 1964), this would result in an Early Holocene fresh-saline interface deepening of ca. 60 m in 100 years (Fig. 7), assuming the saline water body had a common seawater density of 1,025 kg/m³. Less saline water body at depth would result in a deeper fresh-saline interface. If exposure occurred later, during the





mid-Holocene, when rebound rate decreased to 4.5 mm y$^{-1}$ (Forman et al., 2004), fresh-saline interface

will still deepen at a rate of 18 mm y$^{-1}$ (Fig. 7).

The existence of a mixing zone at the top of the epigenetic permafrost (from 5.5 m below current surface) and of water with Cl$^-$ content 15% that of seawater at 3.5 m below the Early Holocene surface suggest that the marine sediment section at ADE was hardly flushed with meteoric water, which further suggests that permafrost aggradation commenced shortly after emergence above the sea (e.g., Kasprzak et al.,

2020). Indeed, some of the simulated freezing scenarios can clearly cope with the above fresh-saline interface deepening rates (e.g., MAAT of -5.8 °C and WFT of 0 °C, Fig. 7). Moreover, assuming that partial freezing (e.g., 50-25%) can also block flushing, this results in even faster permafrost aggradation (Fig. 7b). However, as permafrost deepens, freezing rate slows down (e.g., Fig. 7), and none of the scenarios can cope with the assumed deepening of the fresh-saline interface, which should result in

flushing of deeper zones.

It is suggested that the key factor in fresh-saline interface fossilization in a continuous permafrost landscape is the permanent freezing of the very shallow permafrost, which hydraulically disconnects the sub-permafrost zone from the surface and prevents recharge of this zone with meteoric water. As shown above (Fig. 6 and 7), freezing of the top 3-5 m can occur within several years even with the relatively

high temperatures of the Early to mid-Holocene (e.g., MAAT of -4 °C or even warmer, Fig. 7), therefore fresh-saline interface could effectively be preserved.







Figure 7. Early and mid-Holocene freezing depth in the first 200 years with corresponding MAAT, -5.8 °C, and -4 °C respectively, and with WFT of 0 °C and -2 °C. (a) 100% freezing, (b) 25% freezing. Also shown are curves of two scenarios of corresponding depths of the fresh-saline water interface, using a 1:40 Ghyben-Hertzberg (G-H) approximation for isostasy of 4.5 and 15 mm y$^{-1}$. See text for more details. (c) and (d) are zoomed in of (a) and (b) respectively for 5 m depth and 50 years. The fast deepening in figure b of MAAT -5.8 °C and WFT 0 °C is due to change in porosity as the freezing front reaches the bedrock. The legend describes all figures.



### 6.3 Permafrost aggradation during the Holocene

Gilbert et al., (2018), suggested that our drilling site at ADE emerged from the sea at 10 to 9 Ka BP and
that the delta front advanced westwards at a rate of 4.4 m y$^{-1}$ prior to 9.2 Ka, which decreased to 0.9 m
y$^{-1}$ during the rest of the Holocene. Considering the relatively high rebound rates during 9 to 8 ka (e.g.,
15-19 mm y$^{-1}$), this suggests that land surface at ADE reached 3-4 meters above sea level and a
topographic gradient of 1-2% towards the sea within 200 years. Assuming groundwater table close to
the surface, this should further result in a good flushing of the subsurface, unless freezing took control.
The observed mixing zone, which reaches the very top of the pre-Late Holocene surface, suggests that
freezing started within just a few years after exposure.

Our simulations suggest that both cryotic conditions (i.e., <0°C) and actual ground ice formation were
achieved very soon after exposure to the atmosphere (Fig. 5, 6 and 7), and that significant freezing
depths of 15-33 m can be achieved with 200 years (Fig. 6 and 7). This is true for both the Early to mid-
Holocene warmer period (Kutzbach & Guetter, 1986; McFarlin et al., 2018; Mangerud and Svendsen,
2018; Park et al., 2019; Kjellman et al., 2020) and for any sub-zero average annual temperature,
regardless the size of the annual fluctuations. We even tested MAAT of +1 °C finding that some freezing
could occur (Not shown), which is a seasonal effect, derived from the different thermal conductivities of
ice and water. Our simulations are in good agreement with Harada & Yoshikawa (1996), who estimated
using 1-D model with a MAGST of -5.7 °C but not totally saturated sediments, that 533 years are needed
to freeze 31.7 m of sediments in Moskuslagoon, slightly to the west of our site, on the Adventfjorden
shore.

Our simulations also show that lower percentage of freezing (e.g., 25%) may deepen freezing and
enhance the permafrost aggradation rate (e.g., Fig. 7b); however, this is not true for lower WFT (e.g., 2
°C) or for relatively high MAAT (e.g. 0 °C, compare Fig. 7b b with 7a), which as mentioned above is due
to the trade-off between latent heat and thermal conductivity differences between ice and liquid water.
Nevertheless, MAAT of 0 °C seems unlikely (e.g., Van der Bilt et al., 2019).

In summary, the simulations suggest that permafrost aggradation could and did occur even during the
Early Holocene (10-8 ka BP), and probably as well during exposure in the mid-Holocene. This is in
agreement with Hornum et al., (2020) for the Early to mid-Holocene cooling 9-8 Ka BP. Our findings of
a frozen fresh-saline interface suggests that the ground at ADE remain frozen during the Holocene





thermal maximum which is in disagreement with Hornum et al., 2020, who suggested that the ground thawed and refreeze about 6.5 Ka BP.

Permafrost dynamics during the Holocene were studied in other permafrost regions. While in some areas
there are records of permafrost degradation and peatland expansion already in the Early Holocene (post deglaciation, e.g., Lenz et al., 2015; Kaufman et al., 2015; Grinter et al., 2018; Li et al,. 2021), cumulated evidence indicates that temperature during this period was highly variable, sometimes higher and sometimes lower than in present days (Kaufman et al., 2015). Nevertheless, it is a common observation that during the Holocene Thermal Maximum (mid-Holocene, 8.2-4.2 ka BP) permafrost has been
degrading and thermokarst peaked (e.g., Lenz et al., 2015; Ulrich et al., 2017; Anderson et al., 2019). Permafrost aggradation resumed post- 6 ka, and mainly during the past 4-3 ka (e.g., Grinter et al., 2018; Treat and Jones, 2018).

The ADE site was free of sea water cover in the Early Holocene, prior to 9.2 ka BP. At that time, an abrupt cooling was described in Svalbard (Mangerud and Svendsen, 2017; van der Bilt et al., 2018,
2019). The presented model results show that the initiation of permafrost and its gradual aggradation is possible under relatively high temperatures (yet MAAT≤0°C) of the mid-Holocene. Christiansen et al. (2013) pointed out that local topographic conditions and winds in Adventdalen can induce lower temperatures at low altitude depressions, which could enhance the permafrost aggradation during the mid-Holocene.

**7. Summary and conclusions**

Land surface at northern territories, including western Svalbard, was rising relatively fast in the Early to mid-Holocene. Accordingly, the preservation of frozen saline water (mixing zone) at depths next to Early to mid-Holocene surface is taken as evidence for fast permafrost aggradation, which could halt the infiltration of fresh meteoric water and the flushing of saline water to the sea. This is despite of the
prevailed relatively warm temperatures during this period.

Our modelling confirms that freezing could progress relatively fast down the exposed Adventdalen sediments, i.e. to 15-25 m within 200 years, even under the presumed mid Holocene temperatures.

The modelling further suggests that permafrost may aggrade even when WFT is slightly lower than MAAT, which is due to the differences in thermal properties between ice and liquid water.

Non complete freezing of the cryogenic pore space could result in faster deepening of the freezing front when MAAT is smaller than WFT or even when it is higher, but in the latter the difference is not large





(e.g., MAAT<0 °C and 0<WFT >-2 °C). However, when MAAT>> WFT (e.g., MAAT= 0 °C and WFT ≤ -2 °C), the presence of liquid water in the pore space and its lower thermal conductivity would result in a halt of permafrost aggradation.

**8. Appendix**

Appendix 1: Porosity analises

Selected simulation results demonstrating the effects of the porosity values on the rate of permafrost formation are presented in figures 8 - 11. In general, higher porosity (i.e., more pore water to freeze) will result in slower permafrost aggradation (Fig. 8 and 10) due to the higher latent heat involved. Nevertheless, with lower WFT (i.e.,
closer to MAAT, Fig. 9) the differences in aggradation rates with different porosity values are small and even negligible (Fig. 9 and 11).

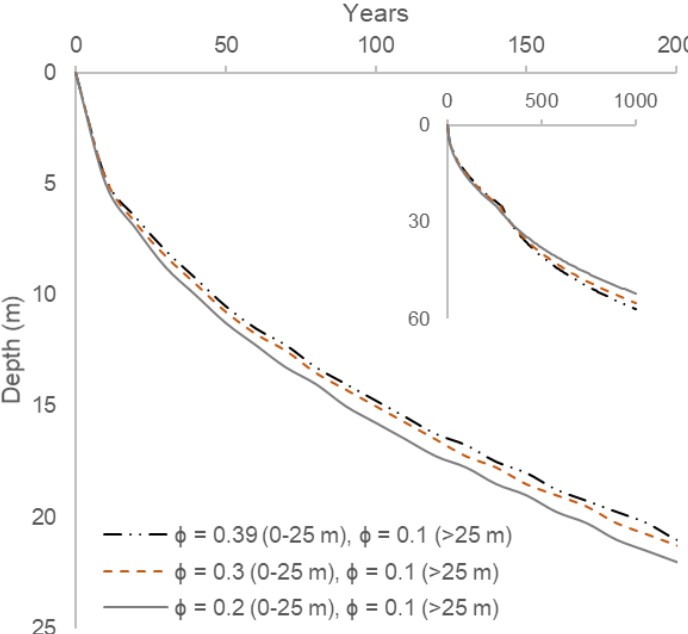

Figure 8. Simulations of freezing front progress with different porosities for MAAT of -4 °C, WFT of 0 °C and 100% freezing. Inset present results for 1000 years. The fast deepening at depth > 25 m is due to change in porosity as
the freezing front reaches the bedrock.



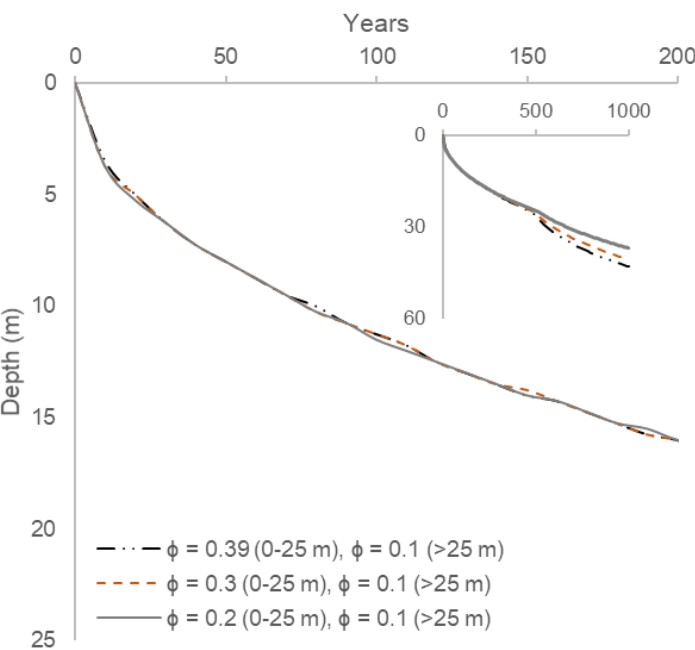

Figure 9. Simulations as in Fig. 8, but with WFT of -2 °C.

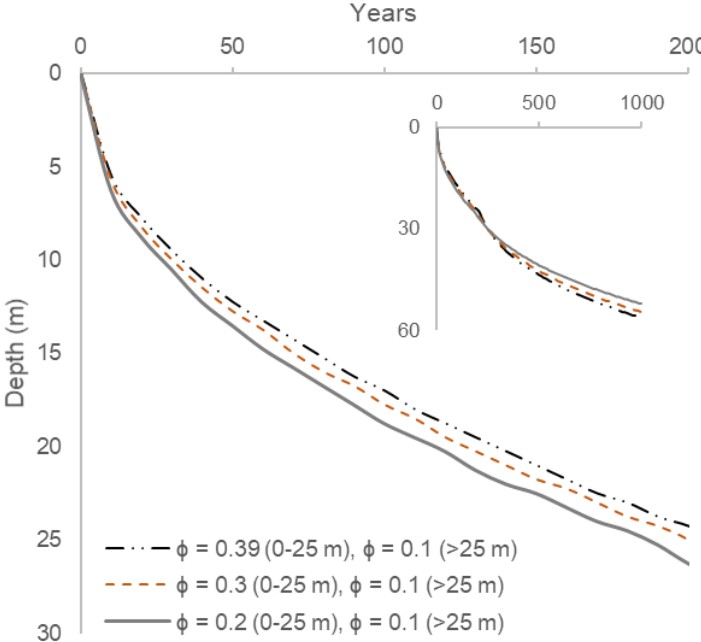

Figure 10. Simulations as in Fig. 8 (MAAT= -4 °C, WFT= 0 °C), but with 25% freezing.





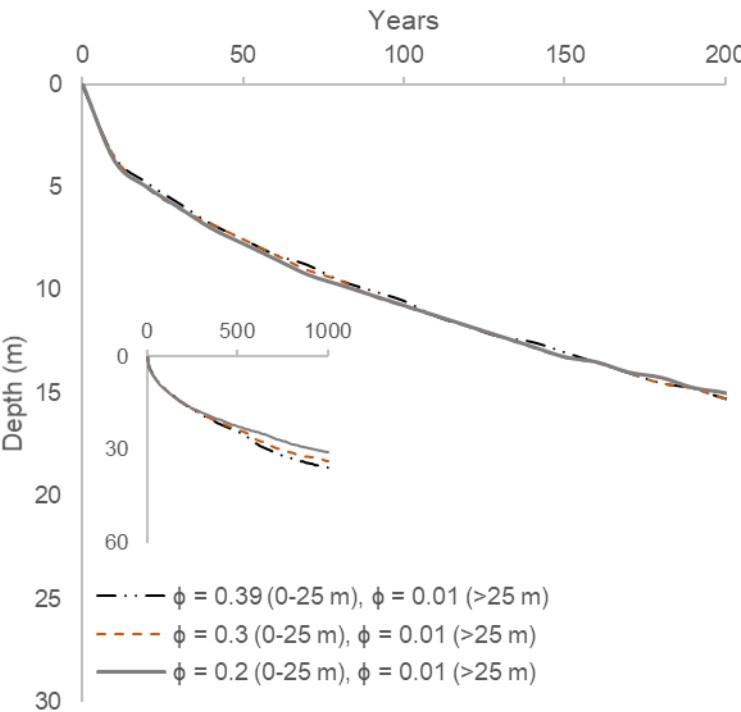

Figure 11. Simulations as in Fig. 9 (MAAT = -4 ᵒC and WFT = -2 ᵒC) but with 25% freezing.

## 9. Code availability

# 1-D freeze/thaw model code – copy and paste code rows into Python (Spider-Anaconda).

# This Python-script describes the 1-D Heat transfer model code developed for the research first Rotem et al., (202….). The 1-D model is a transient one-dimensional heat transfer model suitable for simulating permafrost dynamics. The core of the model is an explicit forward-difference time approximation of the one-dimensional heat transfer equation. This script is tailored to simulate the Holocene ground

temperature development in Adventdalen, Svalbard, but may be modified to fit other purposes.

#Usage must be cited by reference to Rotem et al. (202….).

#For references cited below see:…





```
#Importing relevant packages for Python
import numpy as np
import matplotlib.pyplot as plt
import sys
import pandas as pd

n =1201                  # number of point grid
m1 =5            #number of snapshots – i.e. No. of curves as display in fig. 4 and 5, or No. of sub sets of
data to be exported.
m2 = 365*50              # number of steps for every snapshot
m = m1*m2                # number of time step

dt = 3600.0*24.0/2       #1/6 of a day 14400 sec # 1000000sec =approx. 11.8 days
t_final = dt*m              #defines the final time step
t_days=t_final/3600./24.   #converts time steps to days
t_snap= dt*m2/3600./24.
print ('t_snap')
z=300.0                  #meters depth
dz = (z/(n-1))            #meter - defines the cell width if cell width (dz).
Tfr=0                    #Freezing temperature define as WFT in article

#define depth-porosity profile
Depth = np.zeros(n)
por   = np.zeros(n)
for i in range (0,n):
    Depth [i] = -i*dz
    por[i] = 0.3                 # sediment's porosity
    if Depth[i] < -25.0:       # change from sediment's porosity to rock porosity
      por[i] = 0.1

L = 334000.0              # J/kg water and ice Latent heat
```




```
#Density of materials
p_ice = 916.2          # Kg/m^3
p_water = 999.85     # Kg/m^3
p_soil = 2400.0        # Kg/m^3
```


```
#Defines Heat conductivity (K) & Heat Capacity (Cp) as function of porosity.
K_dufsoil  = 0.35     #W/m*K  K_dufsoil = k of  dry soil at temperatures ca. 5 °C centigrade.
K_dfsoil  = 0.35      #W/m*K K_dfsoil = k of dry soil at temperatures lower than 0 °C
Cp_dufsoil = 837       #heat capacity of dry soil (silt) in ca. 10 °C Cp_dufsoil = Heat capacity of dry soil
```
```
above 0 °C.
Cp_dfsoil = 712         #heat capacity of dry soil (silt) in ca. -10 °C Cp_dfsoil = Heat capacity of dry soil
below 0 °C.
K_ice     = np.zeros(n)
K_water   = np.zeros(n)
```
```
Cp_ice    = np.zeros(n)
Cp_water  = np.zeros(n)

for i in range (0,n):
        K_ice[i]   = 2.24 *por[i] + K_dfsoil *(1-por[i])              # W/m*K   regression with 0<B<1  -1.651x
```
```
+ 2.22
        K_water[i] = 0.569*por[i] + K_dufsoil*(1-por[i])         # W/m*K
        Cp_ice[i]   = 2100.0*por[i] + Cp_dfsoil *(1-por[i])        # j/K*Kg
        Cp_water[i] = 4192.0*por[i] + Cp_dufsoil*(1-por[i])          # j/K*Kg   regression with 0<B<1 2.192x +
2. considering 90% soil and 10% water.
```
```
#control on model stability condition for explicit-in-time numerical scheme
        dd2=K_ice[i]/(p_soil*Cp_ice[i])*dt/dz/dz
        dd3=K_water[i]/(p_soil*Cp_water[i])*dt/dz/dz
        if dd2>0.25 or dd3>0.25:
```
```
                print(' Values must be < 0.25 ')
```



```python
        print(dd2,dd3)
        print(i,Depth[i],por[i])
        sys.exit('  Decrease time step  ')

x=np.linspace(0,z,n)

    #creating time field for data export
    time=np.zeros(n)
    for i in range (0,n):
time[i]=0+dt*i

    #initial conditions
    B=np.ones(n)            # B is a variable between 0 and 1 considering the ratio of ice or water in a cell.
    B=1=water, B=0=ice creates an array of 1
T=np.ones(n)       # Initial Temperature 2 Centigrade creates array of 2 centigrade across the soil profile
    for i in range(0,n-1):
        T[i]= 2.0 + 0.033*i*dz   #Thermal gradient 0.033 centigrade per  m

    #Define working arrays
Tn=np.zeros(n)           # creates an array for each one of the variables
    Bnew=np.zeros(n)
    K=np.zeros(n)
    Cp=np.zeros(n)
    p=np.zeros(n)
dE2=np.zeros(n)

    for j1 in range(0,m1):     #a loop on the snapshots
     for j2 in range(0,m2) :
#Boundary conditions
        Time=(j1*m2+j2)*dt/3600./24/365    #years
```





```
      T[0]=12*np.sin(2.*np.pi*Time)-4   #surface temperature  for seasonal variation

      B[0]=0.      #B=1=water, B=0=ice

      if T[0]>0:

645        B[0]=1

      T[n-1]= 2 + 0.033*z  #bottom of profile temperature. when sediments exposed to air it still has the sea
water temperature.

      B[n-1]=1.    #B=1=water, B=0=ice

for i in range(0,n):      # calculate mixture properties each variable is calculated with linear ratio to B
(ice to water ratio in a cell).

        K[i]=K_ice[i]*(1.0-B[i])+K_water[i]*B[i]

        Cp[i]=Cp_ice[i]*(1.0-B[i])+Cp_water[i]*B[i]

        p[i]=p_ice*(1.0-B[i])+p_water*B[i]


      for i in range(1,n-1):     # loop over internal points. the dE2 (energy equation) equation is split for more
convenient calculations

        dE2[i]=(K[i-1]+K[i])/2.0 * (T[i-1]-T[i])/dz

        dE2[i]=dE2[i]-(K[i]+K[i+1])/2.0 * (T[i]-T[i+1])/dz

dE2[i]=dE2[i]*dt      # Total energy flux into cell "i"

        Tn[i]=T[i]+dE2[i]/(p_soil*Cp[i]*dz)          # conduction calculating the new temperature in the next
cell.

        Bnew[i]=B[i]         #  calculating the new ice/water ratio in the next cell

        if dE2[i]<0.0 and Tn[i]<Tfr:    #condition that verify the amount of energy and the new temperature.
if the energy gets less than 0 value it means that energy is escaping the cell and it will cool down or
freezes.

          if B[i]>0.0001:       # Freezing

            Bnew[i]=B[i]+(dE2[i]-(T[i]-Tfr)*(p[i]*Cp[i]*dz))/(p[i]*L*dz*por[i]) # the  Bnew  depends  on  the
amount of energy that has been used to freeze the previous cell - the rest of the energy

670            if Bnew[i]>0.0: # if the condition is true the new temperature equals freezing temperature.

              Tn[i]=Tfr

            else:
```



```
           Tn[i]=Tfr+Bnew[i]*(L*por[i])/Cp[i]   # if the condition is false (Bnew <0.0) the new temp. equals
     the freezing temp.+Bnew.
675           Bnew[i]=0.

        if dE2[i]>0.0 and Tn[i]>Tfr:

          if B[i]<0.9999:        # Thawing
680           Bnew[i]=B[i]+(dE2[i]-(T[i]-Tfr)*(p[i]*Cp[i]*dz))/(p[i]*L*dz*por[i])
              if Bnew[i]<1.0:
                Tn[i]=Tfr
              else:
                Tn[i]=Tfr+(Bnew[i]-1.0)*(L*por[i])/Cp[i]
685           Bnew[i]=1.

        for i in range(1,n-1):
          T[i]=Tn[i]
          B[i]=Bnew[i]

     plt.plot(T,-x, 'r',label="Temperature",linewidth=0.5)

     #exporting data to csv file. Graphs was created with Microsoft Excel.
      name_dict} =
'       Temperature': T,[:0]
     '       time':time,[:0]
     '       depth': Depth,[:0]
     '       B': B,[:0]
     '       K': K,[:0]
'       Cp': Cp,[:0]
     '       p': p,[:0]
     '       dE2': dE2[:0]
     {
```





```
    df = pd.DataFrame(name_dict)
df.to_csv(r'C:\Users\ADMIN\Desktop\Python        Dotan\1Dmodel_1.csv',mode='a',        header=True,
    float_format='%.3f')  # defines the location of the data exported
    pd.read_csv('trial2.csv').count()

    #Commands for graph in python script. Graphs for article was created with Microsoft Excel.
axes1 = plt.gca()
    plt.ylabel('Depth')
    plt.legend(loc ="lower left")
    plt.grid()
    axes2 = axes1.twiny()
axes1.set_xlabel("Temperature")
    plt.plot(B,-x, 'g',linestyle = '--',linewidth=0.7, label="B")
    axes2.set_xticks([0., .2, .4, .6, .8, 1.0])
    axes2.set_xlabel("B")
    plt.legend(loc ="lower left")
plt.show()
```

## 10. Data availability

All raw data can be provided by the corresponding authors upon request.



## 11. Executable research compendium (ERC)

## 12. Sample availability

## 13. Supplement link: the link to the supplement will be included by Copernicus, if applicable.

## 14. Author contribution:

DR, YW, and HHC planned the drilling campaign; DR and YW, Processed the cores and lab work in UNIS Svalbard; DR and YH preformed the water chemistry analysis at GSI; VL and DR developed the
1-D model; DR, YW and VL wrote the manuscript.

## 15. Competing interests:

The authors declare that they have no conflict of interest.

## 16. Disclaimer

## 17. Acknowledgments

We would like acknowledge Ullrich (Ulli) Neuman for leading the 2017 drilling campaign in Adventdalen. Andy alexander and Graham Lewis Gilbert for field assistance. Danni Rohdent for lab assistance. Gerd-Irena and UNIS logistics for their assistance with field and laboratory gear. GSI geochemical lab members Olga Berlin, Galit Sharabi and Dina Siber for their assistance with chemistry analysis. Yosi Yechieli for consulting about various issues of the article. The drilling campaign was granted by AFG,
Project Number: 269988 RiS ID: 10664.

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
