# Peer review of "Permafrost saline water and Early to Mid-Holocene permafrost aggradation in Svalbard"

_The Cryosphere, 2022_

## Author Response (AR1)

The Cryosphere

Editor

Dear Prof. Hauck,

Please find attached our revised paper

"Permafrost saline water and Early to Mid-Holocene permafrost aggradation in Svalbard".

We would first like to thank you and the two reviewers for the comments to our manuscript.

Following the main comment of reviewer #2 about the thermal properties used in the model, we re-run the model with the suggested thermal conductivity of 3 W $(mK)^{-1}$. This resulted in even faster and deeper permafrost aggradation, confirming and enhancing our suggestion about the freezing of the Fresh Saline Interface (FSI) short time after exposure. We have incorporated and discussed these new results in the revised manuscript. All other comments were carefully taken care of and answered in order in the Reply to Reviewers.

We would appreciate if you consider this manuscript for publication in The cryosphere

Yours sincerely,

Dotan Rotem

We would like to thank and appreciate the comments of the two referees. We paste your comments and commented on each one of them, in bold letters. No. of Lines corresponds the clean version.

**Referee # 1**
The introduction constitutes a very complete assessment of the research background on the studied topics and objectives of the work.
The study area does not provide present- day climate data, which would help to frame to understand (past) simulated conditions.
**- We now added climate data in Lines 102-106.**

The methodological part is very well described, and the different steps of the research approach clearly exposed. Discussion is concise and summarizes the main findings, comparing it with other areas where Holocene permafrost dynamics has been also examined. Conclusions capture also the main findings of the paper.
It would be interesting to add what are the implications of these results for recently exposed (Late Holocene) areas, and how these data can be used to assess on the future evolution of permafrost in Svalbard (aggradation. vs degradation).
**- Thanks for this comment. We added implications to recently exposed permafrost in lines 566-568.**

Specific comments
Page 3, l. 50 – include also snow cover dynamics.
**- we mentioned it in the first version, and now we re-wrote it (Line 54)**

Page 3, l. 52 –warm-based glaciers.
- **Corrected (Line 51).**

Page 3, l. 54 –better to refer to Last Glacial Cycle than to the LGM.
**- Corrected (Line 55).**

Page 4, l. 86 – last glacial cycle
- **Corrected (Line 89).**

Page 4, l. 88 – are all ages calibrated (cal BP)?
- **All ages are cited from papers and they are all BP.**

Page 4, l. 102 – to better understand and frame the study cases presented later in the paper, present-day MAATs should be given here. Similarly, as also described in the Discussion, precipitation values should be included here.
**- present-day MAAT is now described in Lines 103-106 Current and past precipitation is presented in Line 106.**

Consider adding a picture of the study site (and maybe also of the cores) to help the reader better understand the environmental/sedimentological setting.
**- Picture was added to Figure 1b page 6**

**Referee # 2**

Major

1. The time step of 32,600 seconds is not 0.5 days as written in the text. Half a day is 43,200 seconds. Please check the simulations.

**- Thank you for pointing it out. Time step was 10,800 seconds (3 hrs). Corrected in Line 317**

2. I think the thermal conductivity is not correctly calculated and this can have a major impact on the results. In equation 2, the dry soil conductivity is used for the mineral fraction of the soil. However, the dry soil thermal conductivity is a bulk value. In this equation for saturated conditions, the "mineral thermal conductivity" should be used and this is typically around 3.0 W/(mK). Therefore, a value of 0.35 W/ (mK) is excessively low.

**- Thanks for this important comment. A recently published model used a similar thermal conductivity of 0.5 W(mK)$^{-1}$ for the Adventdalen quaternary sediments (Hornum et al., 2020). Nevertheless, we agree that a higher value should also be checked. Accordingly, we re-run the model with conductivity of 3.0 W m$^{-1}$K$^{-1}$. We explain it as a variable in lines 243-250 and we present the results in parantheses in lines 340-365 as well as in in Figure 7 and in the porosity figures 9-12 in the appendix. We discuss these results and the differences between them and those arrived at with 0.35 W m$^{-1}$ K$^{-1}$ in Lines 366-367, 385-388 and 395-403.**

3. Because of the thermal conductivity error, I am skeptical of the permafrost aggradation rates. I am a little bit surprised that changing the porosity has such a small effect on the results, especially because the latent heat associated with such a change is significant. If the authors rerun the simulations with the correct thermal properties, I hypothesize that, the permafrost aggradation rates would be more divergent when considering different porosities.

**- As discussed in Lines 340-365 and 464-491 and shown in Figures 7, 9-12 in most cases, freezing is much faster with the higher conductivity value. This is except for when using very low WFT (see Lines 524-528)**

4. The numerical model does not consider salt diffusion and therefore salts cannot migrate during the advance of the freezing front. While I do not expect the authors to incorporate salt diffusion into their model, I would appreciate some more discussion on this process. As the authors point out, ground freezing results in ionic exclusion, thereby increasing the porewater salt concentration. Consequently, this creates a porewater salt concentration gradient. Since the advance of the freezing front slows with time, the porewater salt concentration can be sufficiently strong at a particular depth to increase the porewater salinity and create a cryopeg or partially frozen conditions. How would the permafrost aggradation rates change if salt transport were included in the model? For coupled heat and salt diffusion models, consider the following paper: https://doi.org/10.1029/2018JF004823.

**We did not include salt diffusion in our model, a process that will reduce WFT as freezing progresses. It can explain the reason for partially frozen samples extracted from the epigenetic section, we added the extracted core condition to table 1. Including salt diffusion, we assume the freezing front may have advanced somewhat slowly than suggested by our model.**
**- We discuss it shortly in lines 493-496.**

5. Please add a conceptual diagram of the Ghyben-Hertzberg approximation and include two panels (1 with permafrost and 1 without permafrost). This would really help the reader visualize how the fresh-saline interface is expected to look in unfrozen and frozen environments.

**The diagram is presented as figure 8 page 26**

6. If available, could you please include ground temperature data with the geochemical ground ice data in Table 1? At the very least, were in-situ frozen and unfrozen conditions recorded during drilling? Please add this information.

**- We did not measure ground temperatures during drilling. We did add a column to table 1. Referring to the frozen state of the retrieved cores.**

Minor:
Line 13: Should "valley" be capitalized?
- **Corrected, (Line 12).**

Line 45: Consider rephrasing to "below 0 °C".
- **Corrected, (Line 45).**

Line 51: Consider pointing out that permafrost can form in taliks beneath lagoons, as well as beneath bottom-fast ice conditions in shallow water. Consider the following paper: Solomon, S. M., Taylor, A. E., & Stevens, C. W. (2008, June). Nearshore ground temperatures, seasonal ice bonding, and permafrost formation within the bottom-fast ice zone, Mackenzie Delta, NWT. In Proceedings of the Ninth International Conference on Permafrost, Fairbanks, Alaska (Vol. 29, pp. 1675-1680). Fairbanks: Institute of Northern Engineering, University of Alaska Fairbanks. **A sentence was added it appears in lines 52-54 and the reference was added to the reference list.**

Line 51: Replace "permafrost usually" with "permafrost is usually". **Corrected, (Line 51).**

Line 54: Replace "Barents Sea" with "the Barents Sea". **Corrected, (Line55).**

Line 68: You mention that groundwater flow is practically impossible in continuous permafrost areas. Can you make a few comments about groundwater flow in cryopegs in continuous permafrost and if this is relevant to Svalbard?.

- **following your comment, several sentences were added including references. It appears in lines 70-74 and the reference was added to the reference list.**

Line 93: Replace "Exposed surface" with "The exposed surface." **Corrected, (Line 96).**

Line 95: Replace "Active layer thickness" with "The active layer thickness." **Corrected, (Line 99).**

Line 103: The units for "km" should not be capitalized. **Corrected, (Line 112).**

Line 104: Replace "Permafrost section" with "The permafrost section." **Corrected, (Line 114).**

Line 105: Replace "1 to 5.5" with "1.0 to 5.5" **Corrected, (Line 115).**

Figure 1: Please improve the resolution and include a higher quality figure.
**It is in high resolution in my computer I embed it in the file in low resolution to avoid large file size. I'll send it to TC.**

Line 118: Replace "with serial" with "with a serial." **Corrected, (Line 134).**

Line 140: Replace "afresh" with "a fresh." **Corrected, (Line 155).**

Line 149: Should "Pingo" be capitalized? **Yes**

Line 153: Please comment on why the high ratio of Ca/Cl and SO4/Cl at a depth of 5.45 m is enigmatic.
**- It may present former active layer - permafrost table zone were elements may concentrate (e.g. Cary and Mayland, 1972; Kokelj et al., 2002). We have added it in the manuscript, thanks (Lines 167-169). Citations are included in the reference list.**

Table 1: Please add a row for "standard seawater composition" to help put the results in context. **We have added it.**

Line 225: Please be consistent. In the text, water freezing temperature (WFT) is used and in some of the figures (e.g., Figure 3) Tf (freezing point) is used. **We have changed it in the diagram and in the text line 198.**

Table 2: Careful with the units of thermal conductivity. The units should be W/ (mK). **We have changed it to W m$^{-1}$K$^{-1}$, Thanks.**

Table 2: Please use appropriate notations for multipliers and exponents. The table looks a little messy. **We have corrected it see table 2.**

Line 320: Please define "winter inflection point."
**- Inflection point is the minimum value shown by the curve deepen with time. The word winter was a mistake and we took it off (Line 345)**

Line 320: Replace "freezing front" with "the freezing front." **Corrected, (Line 346).**

Figure 4: Please add labels for panels "a", "b", and "c." **We have added it, thanks.**

Line 380: Replace "When freezing" with "When the freezing". **Corrected, (Line 425).**

Line 383: What do you mean by "low water activity?"
**- A water activity value of unity indicates pure water, whereas a water activity value of zero indicates the total absence of 'free' water molecules; the addition of solutes consistently lowers the water activity. Nevertheless, we decided to take it out.**

Line 389: For simplicity, why not state the eutectic point of the H2O-NaCl system (-21 °C)? **- - - This is also a possibility, but we prefer to leave it like that.**

Line 427: Replace "Less saline" with "The less saline". **Corrected, the sentence was changed (Line 474).**

Line 428: I suggest replacing "exposure" with "sub-aerial exposure.". **Corrected, (Line 474).**

Line 429: Replace "when rebound" with "when the rebound." **Corrected, Line 475).**

Line 463: Replace "Assuming groundwater" with "Assuming the groundwater." **Corrected, (Line 503).**

---

## Author Response (AR2)

Dear referee 2

Thank you for the comments.

Following the comments, we are now presenting our results only with thermal conductivity of 3 W m$^{-1}$K$^{-1}$.

All other questions and minor corrections were corrected.

**The correction refers to the clean version.**

Dear authors,
Thank you for the review. The line numbers I present here refer to the PDF with tracked changes. I am pleased with most of the responses to my review, except for the thermal conductivity (k) estimations. I would like to better explain my concern in this review round. My comments are divided into major and minor sections.

Major comments:
In the equation 2, theta represents the porosity and B represents the fraction of unfrozen water in the pore space. If B=1, then all the water in unfrozen.
K = (1-theta)*ksoil + theta ((1-B)*kice + B*kwater)
This is basically an equation for saturated conditions. For the mineral fraction we simply subtract theta from 1. The problem is that the authors multiply the mineral fraction by dry soil thermal conductivity. But, the dry soil thermal conductivity is a bulk value that considers air in the pore space. So, the authors cannot use a dry thermal conductivity value for k if the pore space is filled with water and/or ice. Here is a paper by Cosenza with appropriate values for k_mineral:
https://doi.org/10.1046/j.1365-2389.2003.00539.x
This error also applies to the density estimation, because once again, the pore space is saturated. Is p_soil in this case the density of dry soil that accounts for air in the pore space? And depending how you estimated the volumetric heat capacity; this equation would also have a mistake. In general, you need to make sure the values you use in your equation represent saturated conditions. I recommend keeping the simulation results with a mineral thermal conductivity of 3.0 W/mK and removing the results with a mineral thermal conductivity of 0.35 W/mK. If you wish to include a sensitivity analysis, then please use a lower mineral thermal conductivity that makes sense.

**We now changed all to 3 Wm$^{-1}$K$^{-1}$, as suggested.**

My point about the thermal conductivity is also revealed in line 258 of the tracked changes PDF.
Line 258: The thermal conductivity range is defined by dry soil at the low end (a bulk value that includes air) and the mineral conductivity of quartz (8.8 W/mK) at the high end. The thermal conductivity of quartz is the mineral thermal conductivity and not the thermal conductivity of dry sediment.

**We made changes also in this part and cited more literature – lines (248-250)**

Line 375: Is this because the seasonal change in thermal properties is less pronounced when a higher mineral thermal conductivity is applied in the bulk thermal conductivity estimation?

**Yes, the updated, higher conductivity of soil results in enhanced heat loss during summer. Nevertheless, careful examination of simulation results showed that a thin layer (0.5m) may actually formed also with -5°C and 3W $m^{-1}k^{-1}$ (Although not under -6°C). Please see revised text (lines 370-377)**

Line 395: Why is there a small difference between the 100% and 25% freezing conditions for a WFT of -2 degrees Celsius? Is it because the thermal gradient between the WFT and the upper boundary forcing is low?

**Correct. With small difference between the two, partial freezing becomes similar to complete freezing.**

Line 409: Why does increasing the thermal conductivity result in less permafrost aggradation in this case (9.5 m instead of 12 m)?

**Permafrost aggradation depth largely depends on the difference between frozen soil conductivity (winter) and thawed one (summer). The difference between the two is more effective with low dry soil conductivities (0.9 and 0.4, respectively, with the 0.35W $m^{-1}k^{-1}$ for dry soil) than with the high dry soil conductivity of 3W $m^{-1}K^{-1}$ (2.8 and 2.3, respectively). Nevertheless, in the revised version we are left only with the 3W $m^{-1}K^{-1}$.**

Line 520: If the salt diffusion into the sediment produces only partial freezing, then shouldn't the freezing front advance faster? This would be consistent with your partial freezing simulations where you keep the freezing point depression fixed. However, because of the salt build-up via diffusion, saline cryotic unfrozen layers can perhaps develop and affect groundwater flow.

**- Thanks for this question. We basically suggest that freezing of the topmost soil (a few m) inhibits infiltration (also up valley from the ADE site), therefore hinders FSI deepening. We note that at >10m, water does not exceed 15% seawater salinity, which implies freezing temperature quite close to that of fresh water. Also, the preservation of an FSI demonstrates that there was not much of groundwater flow since permafrost formation.**

Minor comments:

Line 14: Replace „cheistry" with „chemistry" – **corrected (line 13)**

Lines 54-56: Please fix the structure of this sentence – **We changed it to:**

**"Nevertheless, permafrost can occur beneath lagoons in association with taliks, as well as beneath bottom-fast ice in shallow water (Solomon et al., 2008)."**

**(Lines 52-53).**

Line 73: What do you mean by "enhancing cryopegs?" i.e. providing the necessary conditions for their existence? **(we changed to "Flow may also provide the necessary conditions for the formation of cryopegs …" (line 71)**

Line 75: Replace "cryopegs was" with "cryopegs were" – **corrected (line 72)**

Line 408: Replace "and Thermal conductivity" with "and a thermal conductivity" – **sentence was erased**

Line 512: Replace "freezing rate" with "the freezing rate." - **corrected line (483)**

Line 593: Replace "was rising" with "were rising". **corrected line (552)**

Figures 5 and 6: Specify the thermal conductivity used. **no longer relevant; all simulation was conducted with 3W m$^{-1}$ K$^{-1}$**

---

## Author Response (AR3)

Dear referee 2

Thank you for the comments and for the efforts you have made to improve our manuscript.

**Please note that line numbers refer to the clean version.**

Main comments:

Table 2:
Why are there unfrozen and frozen values shown for heat capacity and diffusivity for the soil (silt) category? You should only be using one value because your porespace (saturated) is treated separately for temperature and salinity-dependent water/ice fractions. The bulk diffusivity, heat capacity, and thermal conductivity are driven by changes to the water and ice fractions, so specifying unfrozen/frozen values for the mineral fraction is not required. The third row of Table 2 should only be the mineral fraction of the saturated sediment because your equations in (2) weight each component of the sediment (mineral, ice, water) volumetrically. The porespace is also saturated, so air is not considered. This is why you cannot use any "dry soil" value that is derived for a dry soil state with air inside the porespace. It is very important that is cleared up in Table 2 and that there is no confusion. Additional changes to Table 2:
- Please change the title to, "1-D heat transfer model physical parameters of water, ice, and sediment"
- Change "soil (silt)" to "mineral fraction (silt)"
- Show only one value for the mineral fraction's thermal conductivity, heat capacity, and diffusivity

**We have corrected table 2 as proposed**

Minor comments:

- Line 24: Replace "permafrost aggradation" with "The permafrost aggradation"
**corrected (line 23).**
- Line 47: Replace "between atmosphere" with "between the atmosphere"
**corrected (line 47).**
- Line 73: Replace "Tavacoli" with "Tavakoli"
**corrected (line 72).**
- Line 194: Replace "evolving" with "an evolving"
**corrected (line 192).**
- Line 254: Replace "dry sediments" with "the mineral fraction of the sediment",
**corrected (line 251).**
- Line 517: Replace "somewhat slowly" with "somewhat more slowly"
**corrected (line 499).**
- Figure 8: Typo in panel "stage 3": replace "aggrads" with "aggrades"
**corrected, Thanks.**

- Summary and conclusions: This is up to you, but I suggest explaining the seasonality effects when discussing "differences in thermal properties" for your scenarios. If a reader only glances at the abstract and conclusion, a broader explanation here would be helpful.

**Thank you, we have added a conclusion concerning that topic, lines 565-567.**